# Inter-Alpha-Trypsin Inhibitor Heavy Chain 4 Plays an Important Role in the Development and Reproduction of *Nilaparvata lugens*

**DOI:** 10.3390/insects13030303

**Published:** 2022-03-18

**Authors:** Jin-Liang Ji, Shan-Jie Han, Rui-Juan Zhang, Jian-Bin Yu, Ya-Bin Li, Xiao-Ping Yu, Guang-Fu Liu, Yi-Peng Xu

**Affiliations:** Zhejiang Provincial Key Laboratory of Biometrology and Inspection & Quarantine, China Jiliang University, Hangzhou 310018, China; j1793505637@163.com (J.-L.J.); hanshanjie@126.com (S.-J.H.); 18893781226@163.com (R.-J.Z.); y17857687285@163.com (J.-B.Y.); lyb13372418590tt@163.com (Y.-B.L.); yxp@cjlu.edu.cn (X.-P.Y.)

**Keywords:** *Nilaparvata lugens*, inter-alpha-trypsin inhibitor heavy chain 4, RNAi, development, reproduction

## Abstract

**Simple Summary:**

The brown planthopper, *Nilaparvata lugens* (Hemiptera: Delphacidae), is a destructive insect pest of rice. It causes reductions in rice yield and great economic losses. In this study, we used RNAi to explore the function of the inter-alpha-trypsin inhibitor heavy chain 4 (ITIH4) gene in the development and reproduction of the brown planthopper. Our results revealed that *ITIH4* influences the survival, ovarian development, egg production, and egg hatching of this insect, indicating that *ITIH4* plays important roles in development and reproduction. Considering the importance of *ITIH4* in the brown planthopper, it may be a potential target for pest management.

**Abstract:**

The brown planthopper, *Nilaparvata lugens*, is a difficult-to-control insect pest affecting rice yields in Asia. As a structural component of the inter-alpha-trypsin inhibitor (ITI), the inter-alpha-trypsin inhibitor heavy chain (ITIH) has been reported to be involved in various inflammatory or malignant disorders, ovarian development, and ovulation. To reveal the function of ITIH4 in *N. lugens*, the gene encoding *N. lugens* ITIH4 (*NlITIH4*) was cloned and characterized. *Nl*ITIH4 contains a signal peptide, a vault protein inter-alpha-trypsin domain, and a von Willebrand factor type A domain. qPCR analysis showed that *NlITIH4* was expressed at all developmental stages and in all tissues (fat body, ovary, and gut), with the highest expression in the fat body. Double stranded *NlITIH4* (ds*NlITIH4*) injection clearly led to an RNAi-mediated inhibition of the expression of *NlITIH4* and resulted in reduced survival, delayed ovarian development, and reduced egg production and egg hatching. These results indicate that *NlITIH4* plays an important role in the development and reproduction of *N. lugens*.

## 1. Introduction

The brown planthopper (BPH), *Nilaparvata lugens*, is one of the most destructive insect pests affecting rice yields in Asia. It feeds exclusively on rice sap and causes serious losses of rice yields via its spawning activities and by transmitting rice viruses. The application of chemical pesticides and the cultivation of resistant rice varieties are typical measures enacted to control *N. lugens*. However, as a migratory insect, *N. lugens* is an *r*-strategic species with high fecundity, and it also develops strong resistance to pesticides and resistant rice varieties, making effective control of *N. lugens* a difficult endeavor [1,2].

Inter-alpha-trypsin inhibitor family proteins (ITI) have inhibitory activity against trypsin and were initially discovered in serum and urine. They are composed of a common light chain called bikunin and multiple closely related heavy chains (ITIHs) [3,4,5,6]. Bikunin is generally linked to one or two ITIHs, and these complexes form the typical protein-glycosaminoglycan-protein (PGP) structure of ITI [6,7,8]. Bikunin inhibits a wide range of proteases, such as trypsin, granulocyte elastase, plasmin, and cathepsin, and it plays a role in inhibiting urolithiasis, suppressing tumorigenesis and metastasis, and preventing postoperative stress and shock [6,9]. Bikunin has been widely studied, but there is not much research on ITIH. Studies have shown that ITIH is associated with various diseases, including the inflammatory response in local tissues, acute inflammation, and the development of tumors [6]. As the main part of ITI, ITIH was also found to play a significant role alone or in combination with bikunin in subsequent studies [6].

An important function of ITIH involves its ability to esterify with hyaluronic acid (HA) [10,11,12,13]. HA interacts with other extracellular matrix molecules to form a network structure to maintain the stability of the extracellular matrix, and it has an important impact on cell properties, migration, and tumor metastasis [13,14,15]. The ITIH-HA complex is believed to protect joints from inflammatory damage possibly caused by free oxygen radicals in arthritis [16]. In ovarian development and ovum maturation, the construction of the cumulus cell matrix also requires the formation of the ITIH-HA complex in some mammals, including humans, mice, and pigs [17,18,19]. In fact, a study that involved a mouse knockout of the *bikunin* gene showed that ITIH-HA is crucial for ovulation and fertilization [18]. ITIH4 from human and pig plasma has also been confirmed as a substrate for the plasma serine protease kallikrein [17,20]. ITIH4 has been established as an atypical acute phase protein (APP) that can inhibit actin polymerization and phagocytosis of polymorphonuclear cells [21], and it is also associated with various malignant diseases and inflammatory conditions [6,21,22,23]. ITIH4 may function as an APP for the protection of the uterus from the inflammatory response in pig endometrium [17]. Furthermore, ITIH4 has also been extensively studied as a biomarker in cancers and other disorders and in other biological contexts [24,25,26,27,28]. Despite the significance of ITIH4 in mammals, research regarding the function of this protein in insects has not been reported.

In a previous experiment, we found that *N. lugens ITIH4* (*NlITIH4*) was highly expressed in the ovary and that *NlITIH4* was most highly expressed during vitellogenesis, implying that *NlITIH4* may be related to the reproduction of *N. lugens.* Understanding the reproduction of *N. lugens* is essential for its management. Therefore, in the present study, we cloned the *NlITIH4* and analyzed its protein sequence, and then investigated the function of *NlITIH4* in female *N. lugens*.

## 2. Materials and Methods

### 2.1. Experimental Insects

The *N. lugens* population used in this study was originally collected from a rice field in Yuyao (Zhejiang province, China), and the insects were maintained in a climatron at China Jiliang University for 10 years. The insects were reared on rice seedlings (Taichung Native 1, TN1) at 27 ± 1 °C with 75 ± 5% humidity under a 14 h light, 10 h dark photoperiod.

### 2.2. Cloning the cDNA of NlITIH4

Total RNA was extracted from five adult females with a TaKaRa MiniBEST Universal RNA Extraction Kit (Takara, Dalian, China). RNA quality was verified by electrophoresis on a 1% agarose gel and the RNA was quantified with a NanoDrop 2000 spectrophotometer (Thermo Scientific, Waltham, MA, USA). A sample (1 µg) of total RNA was used to synthesize cDNA by removing genomic DNA and then performing reverse transcription with a TaKaRa PrimeScript^TM^ RT reagent Kit with gDNA Eraser (Takara, Dalian, China). The cDNA synthesis reaction consisted of a 10 μL reaction volume (1 μL gDNA Eraser, 2 μL 5× gDNA Eraser Buffer, 1 µg sample RNA and RNase Free dH_2_O) that was incubated at 42 °C for 2 min. Another 10 μL reaction volume, which included 1 μL PrimeScript RT Enzyme Mix I, 2 μL RT Primer Mix, 2 μL Oligo dT Primer, and 1 μL RNase Free dH_2_O, was added, and the mixture was incubated at 37 °C for 15 min and 85 °C for 5 s and cooled to 4 °C.

The synthesized cDNA solution was diluted 10-fold and used as a template for PCR. Based on the *NlITIH4* sequence from our transcriptome database, a specific primer pair was designed by using Primer Premier 5.0 (Table 1), and PCR was performed with TaKaRa Ex Taq (Takara, Dalian, China). The PCR reaction was conducted in a 50 μL reaction volume, which included 0.25 μLTaKaRa Ex Taq, 1 μL forward and reverse primers, 5 μL 10× Ex Taq Buffer, 4 μL dNTP mixture, 1 μL cDNA template, and 39.75 μL sterilized ddH_2_O. The PCR program was: 94 °C for 3 min; followed by 35 cycles of 94 °C for 30 s, 56 °C for 30 s and 72 °C for 3 min; and 72 °C for 10 min. The PCR product was recovered with a TaKaRa MiniBEST Agarose Gel DNA Extraction Kit (Takara, Dalian, China) and cloned into a pMD19-T vector (Takara, Dalian, China). The construct was sequenced by Shanghai Sangon Biological Engineering Technology (Sangon, Shanghai, China).

### 2.3. Sequence Comparison and Bioinformatic Analysis

The protein sequence was translated with Protean, and the tertiary structure was predicted by SWISS-MODEL (https://swissmodel.expasy.org/) (accessed on 30 September 2019). Transmembrane regions, signal peptides, and isoelectric point were predicted by SignalP 4.1 Server (http://www.cbs.dtu.dk/services/SignalP/) (accessed on 30 September 2019). The conserved domain and compound helices were analyzed by SMART (http://smart.embl-heidelberg.de/) (accessed on 30 September 2019). The *Nl*ITIH4 protein sequence was compared with other ITIH4 sequences with NCBI BLAST (https://blast.ncbi.nlm.nih.gov/Blast.cgi) (accessed on 30 September 2019), and similar sequences were aligned with Lasergene MegAlin ClustalW. An evolutionary tree was constructed with the neighbor-joining method (bootstrap = 1000) using MEGA 10.0 software package [29].

### 2.4. Double-Strand RNA Synthesis and Injection

Double-strand RNA (dsRNA) was synthesized in vitro by using T7 RNA polymerase and synthetic cDNA templates. Specific primers containing six protective nucleotide bases and the T7 promoter sequence were designed by Primer Premier 5.0 (Table 1). The dsRNA of *NlITIH4* (ds*NlITIH4*) was synthesized in vitro with the template of *NlITIH4* and the primers *NlITIH4*-dsF and *NlITIH4*-dsR (Table 1). The dsRNA of *GFP* (ds*GFP*) was used for the negative control experiment. The *GFP* gene sequence was synthesized in vitro based on the sequence of the binary vector pCAMBIA-1302 (GenBank: AF234298.1) and was cloned into the pMD19-T vector (Takara, Dalian, China). The ds*GFP* was synthesized in vitro with the template of *GFP* and primers *GFP*-dsF and *GFP*-dsR (Table 1). Then, ds*NlITIH4* and ds*GFP* were synthesized according to the instructions of the MEGAscript^TM^ T7 Transcription Kit (Ambion, Austin, TX, USA). A dsRNA synthesis reaction contained 2 μL ATP solution, 2 μL GTP solution, 2 μL CTP solution, 2 μL UTP solution, 1 μg template DNA, 2 μL Enzyme Mix, and 2 μL 10× Reaction Buffer as well as nuclease-free H_2_O up to 20 μL. The reaction was incubated at 37 °C for 16 h, then 1 μL TURBO DNase was added, followed by incubation at 37 °C for 15 min. The quality and quantity of the RNA were analyzed as mentioned above.

Newly emerged virgin macropterous female adults were briefly placed on ice for anaesthetization. Approximately 0.05 μL 4000 ng/μL dsRNA (approximately 200 ng) was injected into the mesothorax using a manual microinjector. After injection, female adults were mated with male adults at a ratio of 1:2.

### 2.5. Real-Time Quantitative PCR Analysis

The expression of target genes in female *N. lugens* samples was analyzed using real-time quantitative PCR (qPCR), and the experiment was repeated three times. To obtain samples from different developmental stages, five adults (one- to five-days old) and multiple nymphs (1st to 5th instar) that were of similar weights (from 0.011 to 0.013 g) were pooled separately for extraction of total RNA. To obtain samples from different tissues, the ovaries, guts, and fat bodies from 30 female adults were dissected and pooled. To obtain samples from dsRNA-treated insects, the entire bodies of five macropterous female adults at various numbers of days post injection of dsRNA (d.p.i.) were pooled.

RNA extraction and cDNA synthesis of theses samples were performed as described above. Before carrying out qPCR analysis, the synthesized cDNA solution was diluted 10-fold. The qPCR reagent was SYBR^®^ Premix ExTaq™ II (Tli RNaseH Plus) (Takara, Dalian, China). The specific primers for qPCR were designed, and *N. lugens 18S rRNA* (*Nl18S*) was used as the internal reference (Table 1). A reaction system in the amount of 20 µL was used for qPCR with a StepOnePlus™ Real-Time PCR System (ABI, USA). The 20 μL qPCR reaction volume contained 10 μL 2× SYBR Premix Ex TaqPremix Ex Taq II, 0.4 μL 50× ROX Reference Dye, 2 μL cDNA template, 0.8 μL forward and reverse primers, and 6 μL sterilized ddH_2_O, and the reaction program was as follows: 94 °C for 30 s, followed by 40 cycles of 94 °C for 5 s and 60 °C for 30 s. The specificity of primers was detected by melting curve analysis and sequencing, and the relative transcript levels of target genes in different samples were evaluated by the 2^−∆∆Ct^ method [30].

### 2.6. Survival Rate Statistic, Dissection Observation and Fertility Analysis

Thirty female adults in each treatment group were used to assess effects on survival rates. The experiment was repeated three times.

For tissue observations, 10 female adults at different d.p.i. were used for dissection. The females were dissected in phosphate buffer solution under a Nikon SMZ1500 stereozoom microscope (Nikon, Tokyo, Japan) and photographed with the NIS Elements software (Nikon, Tokyo, Japan). The number of ovarioles at different vitellogenic stages was counted.

For fertility analysis, each treated female adult was place with two untreated males in a 50 mL centrifuge tube containing fresh rice seedlings that were changed every two days within the 10 d.p.i., and the replaced rice seedlings were maintained independently in new 50 mL tubes. The number of nymphs newly hatched on the seedlings was counted every day until no nymphs hatched, and finally, the seedlings were dissected under the Nikon SMZ1500 stereozoom microscope to observe and count the unhatched eggs. Fifteen female adults were assessed from the two dsRNA-treated groups, and three repetitions were performed.

### 2.7. Data Analysis

All data were analyzed by tone-way analysis of variance (ANOVA), followed by *Tukey*’s test, using the statistical software package SPSS 22 (IBM, Armonk, NY, USA). Significant differences were considered at *p* < 0.01 or *p* < 0.05, and values are displayed as mean ± SE.

## 3. Results

### 3.1. Identification and Bioinformatic Analysis of NlITIH4

*NlITIH4* cDNA was found to contain an ORF encoding a protein of 907 amino acids, and the sequence was uploaded to NCBI GenBank (GenBank: MH069652). The protein *Nl*ITIH4 was predicted to have a molecular weight of approximately 100.10 kDa and an isoelectric point (pI) of 6.17. As expected, *Nl*ITIH4 contains a signal peptide and two conserved structural domains: a vault protein inter-alpha-trypsin domain (VIT, 64 aa–175 aa) and a von Willebrand factor type A domain (vWFA, 304 aa–517 aa) (Figure 1). ITIH4 widely exists in insects, and a sequence comparison showed that *Nl*ITIH4 had highest similarity (48.56%) to ITIH4 of *Homalodisca vitripennis*, which also belongs to the order Hemiptera with *N. lugens.* A phylogenetic analysis showed that ITIH4 is conserved in Hemiptera insects (Figure 2). The widespread conservation of ITIH4 suggests that it has some basic functions in insects.

### 3.2. Developmental and Tissue-Specific Expression of NlITIH4

A qPCR assay was used to detect the developmental and tissue-specific expression of *NlITIH4*, and our results revealed that *NlITIH4* was expressed in different developmental stages and tissues of the female *N. lugens*. In nymphs, the expression level of *NlITIH4* was highest in the second instar nymphs. In female adults, the expression level of *NlITIH4* increased sharply and reached a peak at 2 days after eclosion, then gradually decreased. Moreover, the level of expression of *NlITIH4* was higher in macropterous than in the brachypterous insects (Figure 3A). The expression of *NlITIH4* in the gut, ovary, and fat body of females was also detected, and the results showed that *NlITIH4* in the fat body had the highest transcript level, followed by the ovary (Figure 3B).

### 3.3. RNAi Effect of dsNlITIH4 Injection

The results of qPCR analyses showed that the expression of *NlITIH4* in adult female insects treated with ds*NlITIH4* was inhibited by 94.31%, 97.99%, 99.68%, 99.90% and 99.16% at 1 d.p.i., 2 d.p.i., 3 d.p.i., 4 d.p.i. and 5 d.p.i., respectively (Figure 4A), indicating that the ds*NlITIH4* RNAi was effective. In addition, the survival of *N. lugens* was markedly affected by the ds*NlITIH4* injection. The ds*NlITIH4*-treated group had a lower survival rate than did the ds*GFP*-treated group (Figure 4B). At 1 d.p.i., the survival rates of the ds*NlITIH4*-treated group and the ds*GFP*-treated group were 21.36% and 92.40%, respectively. At 4 d.p.i., the survival rates of the ds*NlITIH4*-treated group and the ds*GFP*-treated group were 12.69% and 67.50%, respectively.

The rate of egg production and the hatching rate of eggs were also reduced by the ds*NlITIH4* injection. We counted the number of eggs at 10 d.p.i. to determine the egg-hatching rate. After ds*NlITIH4* treatment, 30 eggs on average were laid by each female, and these eggs had a hatching rate of 63.5%. In this group, the largest number of eggs laid by an individual was 42, and these eggs produced 25 hatched nymphs. After ds*GFP* treatment, each female laid 90 eggs on average with a hatching rate of 87.7%, and the largest number of eggs laid was 180 with 162 hatched nymphs (Figure 4C).

We furthermore dissected the dsRNA-treated female adults to observe their ovaries, and we found that the vitellogenesis of ovarioles was delayed by the ds*NlITIH4* injection. At 2 d.p.i., approximately 90% of ovarioles in ds*GFP*-treated female adults progressed to late or terminal vitellogenesis, but less than 70% of ovarioles in ds*NlITIH4*-treated did so (Figure 4D). The ovarian development of female *N. lugens* was also suppressed by the ds*NlITIH4* injection (Figure 5). At 2 d.p.i., most ovaries had no visible differences between ds*NlITIH4* and ds*GFP* injection, but a morphological difference was observable from 3 d.p.i. At 3 d.p.i., in the ds*GFP*-treated group, the ovaries were normal with many banana-shaped mature eggs, but in the ds*NlITIH4*-treated group, the ovarian development was obviously delayed, as more incompletely developed eggs were present than in the ds*GFP* group (Figure 5).

### 3.4. Effect of RNAi on the Expression of Vg, VgR and MeT

As described above, RNAi exerted by ds*NlITIH4* caused adverse effects on the ovarian development of *N. lugens*. In the ovarian development in *N. lugens*, the yolk protein accumulates upon the elevated expression of the gene encoding vitellogenin *(Vg)* after eclosion. We therefore further analyzed the expression of *Vg* and the gene encoding the vitellogenin receptor (*VgR*) [31]. We found that the expression of *Vg* was significantly downregulated after the ds*NlITIH4* injection as compared with the expression in control insects treated with ds*GFP* (Figure 6A), indicating that the arrested ovarian development of *N. lugens* upon knockdown of *NlITIH4* expression may be due to decreased *Vg* expression. Similarly, the expression of *VgR* was also considerably reduced after the ds*NlITIH4* injection as compared with the control (Figure 6B). These results indicate that *NlITIH4* may affect the ovarian development of *N. lugens* by regulating the expression of *Vg* and *VgR*. We also examined the relative expression of the gene encoding methoprene-tolerant *(MeT*) from 1 to 4 d.p.i. Over this time, the expression of *Met* was lowest in both groups at 3 d.p.i. It was upregulated by ds*NlITIH4* at 2 and 4 d.p.i. but downregulated at 3 d.p.i. (Figure 6C).

## 4. Discussion

In the present study, *NlITIH4* was found to be expressed in all developmental stages, and in the fat bodies, ovaries and guts of *N. lugens*, and a ds*NlITIH4* injection notably inhibited the expression of *NlITIH4* in the whole body. This inhibition lowered the survival rate of *N. lugens*. These results indicate that *Nl*ITIH4 is important to the growth and development of *N. lugens*. This importance might be due to the following reasons. Firstly, the silencing of *NlITIH4* may have led to the vWFA domain undergoing destruction, or the binding of key molecules that interact with *Nl*ITIH4, such as the ITIH4-HA complex, to be affected. This may have induced a decrease in the stability of the extracellular matrix (ECM) and a loss in function of the key molecules, which in turn would cause systemic function damage. *Nl*ITIH4 in this study was found to be carrying a vWFA domain that has multiple interaction surfaces that can bind to various molecules and that are involved in regulating protein–protein interactions of ECM components [32]. ITIH is also involved in maintaining the stability of ECM by binding to HA [19]. In addition, after *NlITIH4* expression was inhibited, the immune system of *N. lugens* may have been altered, because as an acute phase protein (APP), *Nl*ITIH4 may play a role in tissue injury, the response to stressors acting on the body, and non-specific immunity [21,33].

The effect of ds*NlITIH4* RNAi on the ovarian development and reproduction of *N. lugens* was also significant. After RNAi, the ovarian development was delayed, and the production and hatching rate of eggs were reduced. The possible reasons for these results may be explained as follows.

First, after *NlITIH4* was silenced, yolk protein synthesis was blocked and vitellogenesis was delayed, leading to aberrant ovarian development. This alteration, in turn, would be expected to influence egg-laying capacity and egg-hatching. Vitellogenin and its receptor VgR play an important role in the synthesis of yolk protein and vitellogenesis [31,34]. Lu et al. [31] showed, for instance, that vitellogenin and the VgR were crucial for oocyte maturation and the reproduction of *N. lugens*, and ds*NlVg* RNAi caused oviposition failure. In the present study, the expression of *Vg* and *VgR* was considerably reduced by ds*NlITIH4* RNAi. This mechanism then explains, to some extent, why *N. lugens* reproduction was affected by ds*NlITIH4* RNAi.

In addition, *Nl*Vg, the precursor of yolk protein, is synthesized primarily in the fat body, then is transported through the hemolymph of the circulatory system and absorbed into the ovary by *Nl*VgR-regulated endocytosis. It is ultimately deposited in the ovary, and then the yolk protein is formed [35,36,37]. Here, we found that *NlITIH4* was most highly expressed in the fat body and ovary. This indicates that there may be some connection or interaction between *Nl*ITIH4 and *Nl*Vg and *Nl*VgR. The silencing of VgR could disrupt the uptake of Vg into the developing oocyte, and it may result in the accumulation of Vg in hemolymph; however, ds*NlVg* RNAi had no effect on *Nl*VgR [31]. Since ds*NlITIH4* RNAi decreased the expression of *NlVg**R*, *Nl*Vg content would be decreased in the ovary and accumulated in the hemolymph.

The Knirps-related nuclear receptor (KNRL) controls vitellin (Vn) breakdown in embryos via the transcriptional inhibition of hydrolases in *N. lugens* [38]. In previous work, the expression levels of five selected trypsin genes and the enzymatic activities of trypsin in embryos were found to be significantly increased after KNRL knockdown, and trypsin injection prolonged egg duration, delayed embryonic development, accelerated Vn breakdown, and severely reduced egg hatchability, a pattern similar to that observed in KNRL-silenced insects [38]. Thus it is also possible that silencing of *Nl*ITIH4 may increasingly affect *Vg* expression through an impact on trypsin activity.

The Juvenile hormone (JH) can stimulate vitellogenesis by activating the synthesis and uptake of Vg in *N. lugens*; this may serve as the regulatory basis for oocyte maturation [31,39]. Methoprene-tolerant (MeT) is the receptor for JH and is required for the ovarian development of *N. lugens* [40,41]. In the present study, we detected the expression of *N. lugens MeT* (*NlMet*) after ds*NlITIH4* RNAi. However, the pattern of expression of *NlMet* was not like that of *NlVg* or *NlVgR.* The expression of *NlMet* was downregulated by the ds*NlITIH4* injection only at 3 d.p.i., whereas it was upregulated at 2 and 4 d.p.i.. Similar to the results reported in a study by Lin et al. [41], the expression of *NlMet* decreases when the ovaries of *N. lugens* become mature. The reason why the pattern of expression of *NlMet* was not consistent with that of *Vg* may be due to some feedback regulations in the JH signaling pathway that needs further investigation.

Notably, if ds*NlITIH4* RNAi destroyed the interaction between *Nl*ITIH4 and HA (ITIH4-HA), then the stability of the HA matrix would decrease, and follicular cells or oocyte complex ECM expansion would be blocked. These factors would be expected to impede the maturation of oocytes, fertilization or ovulation, and thus cause the reduction of egg production and hatching rate of *N. lugens* that was observed. Cumulus cells and oocyte complexes and their expansion play an important role in oocyte fertilization, maturation, and ovulation in mammals [42,43,44,45,46]. The formation of the ITIH-HA complex is critical to the construction of the cumulus ECM, and it acts on cumulus oocyte complex expansion, oocyte maturation, fertilization, and ovulation [6,17,18,19]. A study by Fülöp et al. [47] also showed that the impairment of ITI synthesis or HA binding activity could produce unstable HA and hinder oocyte complex expansion and ovulation. Obayashi et al. [48] showed that the inhibition of ITI-HA binding led to ovulation disorder and loss of the fertilization capacity of oocytes. These results suggest that the *Nl*ITIH4-HA complex is likely to be destroyed after ds*NlITIH4* RNAi, and it would ultimately affect the ovarian development and egg production of *N. lugens*. In the future, the effects of the ITIH4 and HA binding mechanisms on ovarian development, and the function of *Nl*ITIH4 for reproduction of male *N. lugens*, will be further explored.

The assembly of ovarian muscle ECM may also be affected, and this would cause the decline of ovarian muscle contractility that externally generates a mechanical force to promote ovulation, resulting in difficult ovulation [40]. Apart from these potential factors, the hydrolysis activities of some proteases, the inflammatory reactions, and immunity related to follicle rupture may also be affected by RNAi. As an APP protein, the disruption of *Nl*ITIH4 and the protease inhibitor activity of *Nl*ITI after ds*NlITIH4* injection would result in the failure of follicle rupture, which in turn would cause a block to ovulation.

## Figures and Tables

**Figure 1 insects-13-00303-f001:**
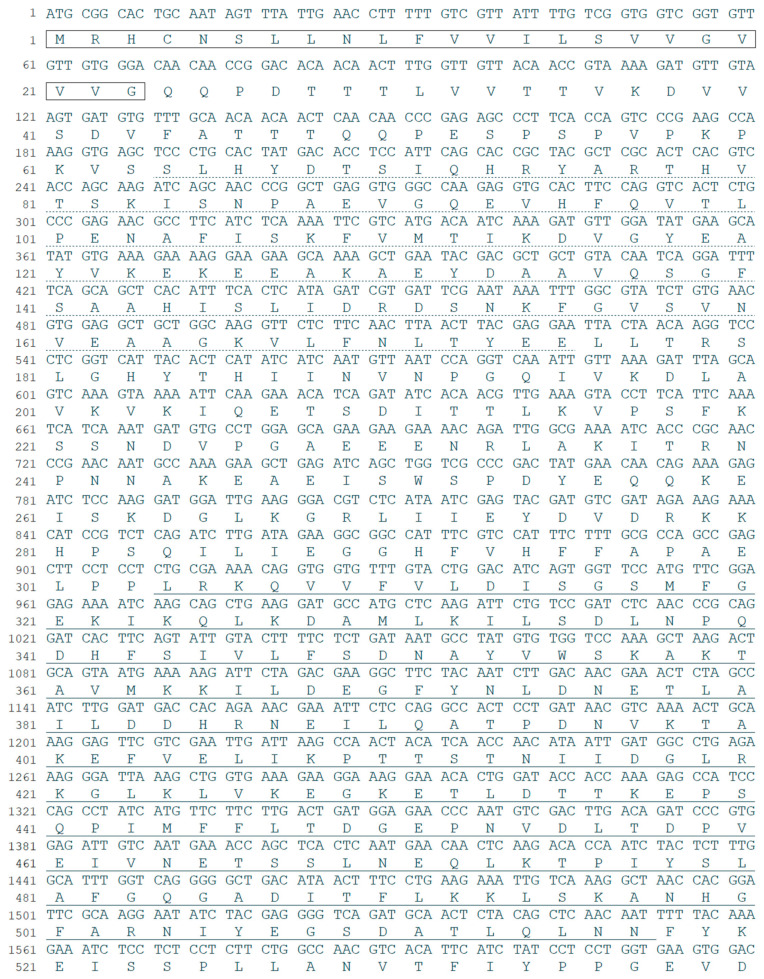
Nucleotide and encoded amino acid sequences of *NlITIH4* ORF. The signal peptide is indicated by a box. The vault protein inter-alpha-trypsin domain is underlined with a dotted line, and the von Willebrand factor type A domain is underlined with a solid line.

**Figure 2 insects-13-00303-f002:**
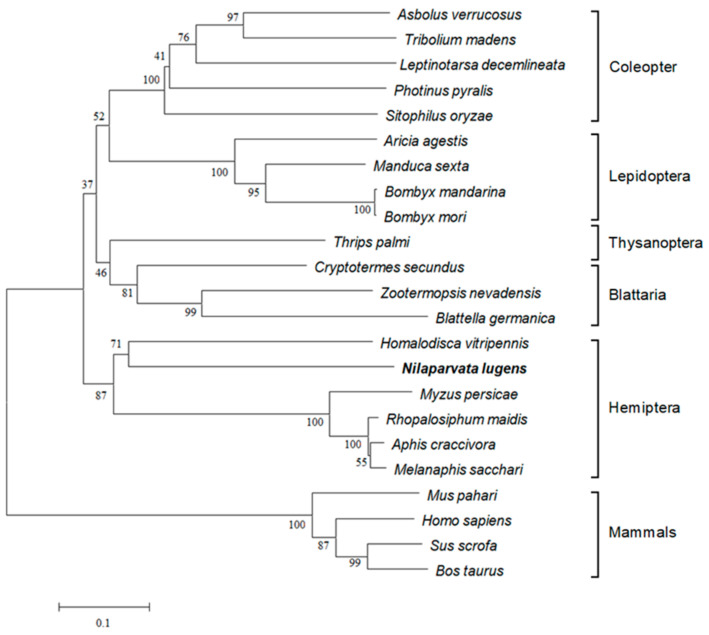
A phylogenetic analysis of ITIH4 based on amino acid sequence. Sequences were retrieved from the NCBI protein database. These sequences included the mammal ITIH4s from *Homo sapiens* (NP_001159921.1), *Sus scrofa* (XP_020924178.1), *Bos taurus* (XP_024838404.1), *Mus pahari* (XP_029397816.1), and all available insect ITIH4s from *Aricia agestis* (XP_041974567.1), *Bombyx mandarina* (XP_028042060.1), *Manduca sexta* (XP_030027688.2), *B. mori* (XP_021202615.2), *Sitophilus oryzae* (XP_030759847.1), *Asbolus verrucosus* (RZB41582.1), *Tribolium madens* (XP_044254677.1), *Leptinotarsa decemlineata* (XP_023015426.1), *Photinus pyralis* (XP_031349540.1), *Homalodisca vitripennis* (XP_046679415.1), *Rhopalosiphum maidis* (XP_026822307.1), *Aphis craccivora* (KAF0720343.1, *Melanaphis sacchari* (XP_025205475.1), *Myzus persicae* (XP_022172078.1), *Nilaparvata lugens* (QCI55997.1), *Cryptotermes secundus* (XP_023714342.1), *Zootermopsis nevadensis* (XP_021930042.1), *Blattella germanica* (PSN48673.1), and *Thrips palmi* (XP_034239235.1). The phylogenetic tree was constructed with the neighbor-joining method (bootstrap = 1000) using MEGA 10.0.

**Figure 3 insects-13-00303-f003:**
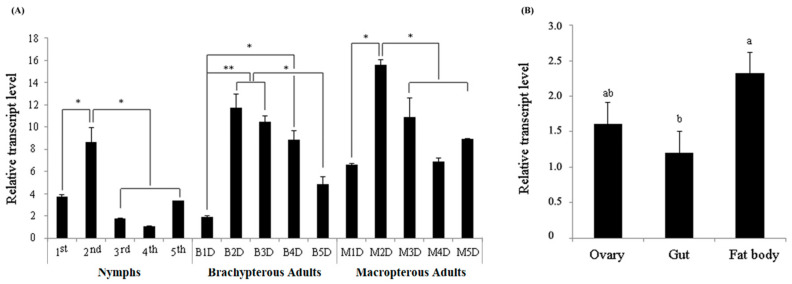
Expression patterns of *Nl**ITIH4* in different developmental stages and tissues. (**A**) Expression of *Nl**ITIH4* in different developmental stages, including 1st to 5th instar nymphs, one- to five-days old brachypterous adult females (B1D to B5D) and macropterous adult females (M1D to M5D). (**B**) Expression of *NlITIH4* in ovary, gut, and fat body of female adults. Data are expressed as mean ± SE. Significant differences: * *p* < 0.05, ** *p* < 0.01. Different lower-case letters above the bars indicate significant differences for which *p* < 0.05.

**Figure 4 insects-13-00303-f004:**
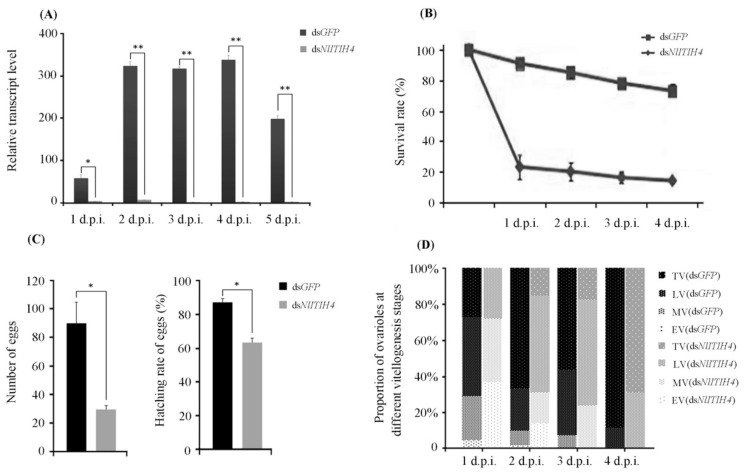
Effects of ds*NlITIH4* injection on the growth and reproduction of *N. lugens*: (**A**) the efficiency of RNAi by ds*NlITIH4* injection at 1 to 5 days post injection of dsRNA (d.p.i.); (**B**) effect of ds*NlITIH4* injection on the survival of *N. lugens* at 1 to 4 d.p.i.; (**C**) effect of ds*NlITIH4* injection on the number and hatching rate of eggs; and (**D**) effect of ds*NlITIH4* injection on the vitellogenesis of ovarioles. EV (early vitellogenesis prophase), MV (middle vitellogenesis), LV (late vitellogenesis), and TV (terminal vitellogenesis) at 1 to 4 d.p.i. Data are expressed as mean ± SE. Significant differences: * *p* < 0.05, ** *p* < 0.01.

**Figure 5 insects-13-00303-f005:**
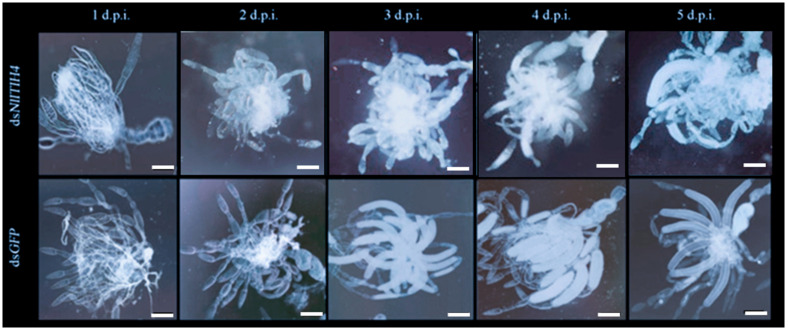
Effects of dsRNA injection on ovarian development at 1 to 5 days post injection of dsRNA (d.p.i.). At 1 to 2 d.p.i., there was no visible difference between the ovaries from ds*NlITIH4*-treated and ds*GFP*-treated groups, but notable differences are present at 3 to 5 d.p.i. Scale bar: 400 μm.

**Figure 6 insects-13-00303-f006:**
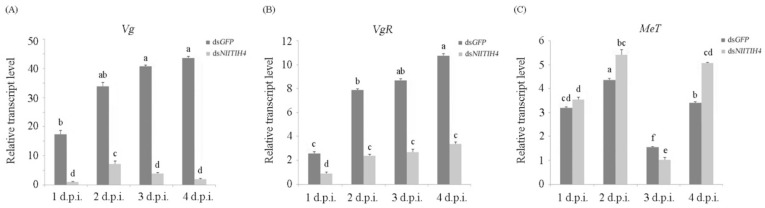
Effects of RNAi on the transcription levels of *Vg* (**A**), *VgR* (**B**), *MeT* (**C**) at 1 to 4 days post injection of dsRNA (d.p.i.). Data are expressed as mean ± SE. Significant difference: different lower-case letters above the bars indicate significant differences for which *p* < 0.05.

**Table 1 insects-13-00303-t001:** The primers used in this study.

Primers	Primer Sequence (5′–3′)	Product Length
For cDNA cloning:		
*NlITIH4*-F	TTTTGCTAACATTTTCCTCCTTG	2798 bp
*NlITIH4*-R	AGTTTCTACGGCTTACTCATCAC	
For qPCR:		
*NlITIH4*-qF	AAGAAAAGGAAGAAGCAAAAGC	191 bp
*NlITIH4*-qR	ATGAGTGTAATGACCGAGGGA	
*Nl18S*-qF	GTAACCCGCTGAACCTCC	170 bp
*Nl18S*-qR	GTCCGAAGACCTCACTAAATCA	
*Vg*-qF	TTCCGTTTGCAGCCACCTATG	154 bp
*Vg*-qR	CTGCTGCTGCTGCTTCTGTCA	
*VgR*-qF	AGGCAGCCACACAGATAACCGC	136 bp
*VgR*-qR	AGCCGCTCGCTCCAGAACATT	
*Met*-qF	GGTGGTAAACGGATTGGAAA	100 bp
*Met*-qR	CATCGTCAGCCAACTCGATA	
For dsRNA synthesis:		
*NlITIH4*-dsF	GGATCCTAATACGACTCACTATAGGACATCAGTGGTTCCATGT	488 bp
*NlITIH4*-dsR	GGATCCTAATACGACTCACTATAGGGATCTGTCAAGTCGACA	
*GFP*-dsF	GGATCCTAATACGACTCACTATAGGGATACGTGCAGGAGAGGAC	350 bp
*GFP*-dsR	GGATCCTAATACGACTCACTATAGGGCAGATTGTGTGGACAGG	

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
