# Peer review of "Inter-Alpha-Trypsin Inhibitor Heavy Chain 4 Plays an Important Role in the Development and Reproduction of Nilaparvata lugens"

_insects, 2022, doi:10.3390/insects13030303_

Round 1

Reviewer 1 Report

Inter-alpha-trypsin Inhibitor Heavy Chain 4 Plays A Role in the Reproduction and Development of Nilaparvata lugens is an interesting work about the insect reproductive system. 

However, the methods and results need to be improved. 

  • First, no description of how many insects, biological replicates, experiment repetitions were described.
  • Despite inform that "Total RNA was extracted from five adult females" no description about nymphs and tissues RNA extraction was reported (how many nymphs/tissue were used per instar?);
  • No description of PCR protocol for amplification of NlITIH4 gene nor dsRNA synthesis was informed: cDNA concentration, buffer, primers, enzyme volume, thermal profile, etc.
  • Was the cDNA diluted prior to the qPCR reaction?
  • In line 155 the authors indicate a Neighbor-joining three but do not describe how it was done nor the three figure. 
  • Figure 1: is it relative gene expression or normalized gene expression? If it is a relative gene expression, the authors need to indicate which sample those are relative to. 
  • Figure 2b: there are no statistics. The sentence "three groups, 30 individuals in each group" is confusing. Are these biological replicates? 
  • Figure 3: The ovarian photos don't look like they were taken under the same conditions. Scale bars are missing. 
  • In line 217, the authors wrote "For ovarian development, the yolk protein is accumulated with the elevated expression of Vitellogenin (Vg) after eclosion of female N. lugens, we therefore further analyzed the expression of Vg and Vitellogenin Receptor (VgR)" it is not clear if this is a result or a citation. 
  • Several sentences along the text are very confusing, specially on results and discussion sections, I recommend a writing revision. 

Author Response

Thank you for your comments and suggestions !

First, no description of how many insects, biological replicates, experiment repetitions were described.Despite inform that "Total RNA was extracted from five adult females" no description about nymphs and tissues RNA extraction was reported (how many nymphs/tissue were used per instar?); No description of PCR protocol for amplification of NlITIH4 gene nor dsRNA synthesis was informed: cDNA concentration, buffer, primers, enzyme volume, thermal profile, etc.

Response: We are extremely grateful to you for pointing out this problem. We have added more details in the “2. Materials and Methods”.

Was the cDNA diluted prior to the qPCR reaction?

Response : Thank you for your careful review. The cDNA solution was diluted 10-fold for the qPCR reaction, and we have added this description in the text (Part 2.5 ).

In line 155 the authors indicate a Neighbor-joining three but do not describe how it was done nor the three figure. 

Response : Thank you for your comments. We have added a figure of the phylogenetic tree, and have descripted the method in the text (Part 2.3),“An evolutionary tree was constructed with the neighbor-joining method (bootstrap=1000) using MEGA 10.0 software package.”

Figure 1: is it relative gene expression or normalized gene expression? If it is a relative gene expression, the authors need to indicate which sample those are relative to. 

Response : Thank you for your comment. According to 2−∆∆Ct method (Livak,K.J.; Schmittgen,T.D. Analysis of Relative Gene Expression Data Using Real-Time Quantitative PCR and the 2−ΔΔCT Method. Methods. 2001, 25, 402-408. doi:10.1006/meth.2001.1262), when we did relative expression analysis, we took the 18S as the reference gene, and set the expression level of the sample who had the highest value of [CT(target gene) - CT(18S)] as 1.

Figure 2b: there are no statistics. The sentence "three groups, 30 individuals in each group" is confusing. Are these biological replicates? 

Response : Yes. Thanks for reminding us. They are “biological replicates”. We have deleted these in the legend, but we add the information in the “2.6. Survival Rate Statistic, Dissection Observation and Fertility Analysis”.

Figure 3: The ovarian photos don't look like they were taken under the same conditions. Scale bars are missing. 

Response : We are extremely grateful to you for pointing out this problem. We have done RNAi experiment again and photographed new pictures. We also have added scale bars in the photos (Figure 5).

In line 217, the authors wrote "For ovarian development, the yolk protein is accumulated with the elevated expression of Vitellogenin (Vg) after eclosion of female N. lugens, we therefore further analyzed the expression of Vg and Vitellogenin Receptor (VgR)" it is not clear if this is a result or a citation. 

Response : It is a citation. We have added a citation (Lu et al,.2015) to the sentence in the text (Part 3.4, line 291).

Several sentences along the text are very confusing, specially on results and discussion sections, I recommend a writing revision. 

Response : We have performed a writing revision through a professional copyediting service. We hope the revised manuscript can be more clearly understood.

Reviewer 2 Report

This study has explored the function of ITIH4 gene in the development and reproduction of Nilaparvata lugens with RNA interference. I think it is relatively simple, and some comments have been made.

  1. Line 79-81, how many generations the insects were kept at lab?
  2. Line 104, can males express the NlITIH4?
  3. Line 112, please add reference (Livak and Schmittgen 2001). 
  4. Line 135-136, will the replaced rice wither before the N.lugens hatches? This has a critical impact on the hatchability data. 
  5. Line 131-142, how many biological replicates were set for survival, spawning, and hatching experiments? and how many individuals were dissected for ovary observation? 
  6. Line 132, which wing type was used for fertility analysis? 
  7. Figure 2C, difference significance analysis between the two treatment groups should be marked. In addition, hatching rate should be calculated and showed. 
  8. Line 193-196 (Fig 2D), is there any biological duplication of egg maturation statistics? If yes, please mark the standard error in the diagram. 
  9. Fig 3, the parameters used in each image were inconsistent, somewhat affecting the comparison between experimental group and control group. 
  10. Line 213, figure legend can be more detailed there. 
  11. Line 216-219, which tissue was used to detect Vg and VgR expression? please describe it at Materials and Methods. 
  12. Line 240, delete “And”. 
  13. You can ask colleagues who are proficient in English for helping to improve the use of English in this manuscript.

Author Response

Thank you for your comments and suggestions !

  1. Line 79-81, how many generations the insects were kept at lab?

Response 1: Thank you for your comments. The insects have been kept at lab for 10 years, and this relevant information has been added in the text (Part 2.1, line 84-85).

  1. Line 104, can males express the NlITIH4?

Response 2: Thank you for your comments. From the beginning of this study, we focused on the function of ITIH4 in reproduction, so, about adults, we only detected females’ expression. In males, NlITIH4 can be expressed.

  1. Line 112, please add reference (Livak and Schmittgen 2001). 

Response 3: Thanks!  We have added the reference to the sentence in the text (Part 2.5, line 167).

  1. Line 135-136, will the replaced rice wither before the N. lugens hatches? This has a critical impact on the hatchability data. 

Response 4: Within 10 days, the leaf blade (with little or no eggs) of some rice seedlings withered, but the leaf sheath (where most eggs are laid) did not completely wither. During experiment, seedlings were maintained in environment with high humidity (70~80%). So, this did not have a critical impact on the hatchability.

  1. Line 131-142, how many biological replicates were set for survival, spawning, and hatching experiments? and how many individuals were dissected for ovary observation? 

Response 5: Thank you for your careful review. Three biological replicates were respectively set for survival, spawning, and hatching experiments. And 10 female adults were dissected for ovary observation. We have added these more information in the “Materials and Methods ”(Part 2.6, line 170-184).

  1. Line 132, which wing type was used for fertility analysis? 

Response 6: Thank you for your careful review. The macropterous female adults were used for fertility analysis, and we have added this description in the “Materials and Methods ” (Part 2.4, line 140).

  1. Figure 2C, difference significance analysis between the two treatment groups should be marked. In addition, hatching rate should be calculated and showed. 

Response 7: We are grateful for the suggestion. We have revised the Figure 4C (old Figure 2C).

  1. Line 193-196 (Fig 2D), is there any biological duplication of egg maturation statistics? If yes, please mark the standard error in the diagram. 

Response 8: Thanks for your comments. There are biological repetitions. For better showing the proportion of ovarioles at different vitellogenesis stage, we presented the data like this, but at the expense of marking the significance.

  1. Fig 3, the parameters used in each image were inconsistent, somewhat affecting the comparison between experimental group and control group. 

Response 9: Thank you for your comments. We have done RNAi experiment again and photographed new pictures. We also have added scale bars in the photos.

  1. Line 213, figure legend can be more detailed there. 

Response 10: We are grateful for your suggestion. We have added more descriptions there (Line 282).

  1. Line 216-219, which tissue was used to detect Vg and VgR expression? please describe it at Materials and Methods. 

Response 11: Thank you for your careful review and suggestion. The whole body of insect was used to detect Vg and VgR expression. And we have described it in “Materials and Methods” (Part 2.5, line 152-154). 

  1. Line 240, delete “And”. 

Response 12: Thanks. We have deleted “And” in the text (Line 319). And we have revised this paragraph.

  1. You can ask colleagues who are proficient in English for helping to improve the use of English in this manuscript.

Response 13: We have performed a writing revision through a professional copyediting service. We hope the revised manuscript can be more clearly understood.

Reviewer 3 Report

In this manuscript, Ji et al. used the typical methodology of molecular biology  to unveil the functional effects of ITIH in N. lugens, which gives us some interesting data regarding the research fields of insect physiology. I believe this manuscript could be a sort of attractive to the related research society. However, there are still many visible issues exist. So a major revision is reasonable for this version. 

Major:

Line 39: Authors used a whole paragraph to make a brief introducing of N. lugens, and the conclusion of this paragraph is "These make it difficult to effectively control N. lugens", which is a big confusion to me. Because I could not find any clues on pest control aspects in this manuscript.

Line 68: Authors mentioned that "very little research has been done on ITIH in insects" which also means that there could still be a few studies referring to this gene. If yes, I strongly recommend listing all the related insect studies. If not, please make a proper change to this sentence.

Line 69-76: The entire paragraph is a big mess, please re-write this paragraph and make a brief review on your past works. And you need to quote the articles if you have already published those data. And if the data haven't been published, I guess you could easily make a combination with the data here. 

Line 79-81: Please provide more information on the insect background and the rearing methods.

Part 2.2: How did you get the full CDS information of the NlITIH4? By conducting RACE PCR or directly deriving from the genome database?

Table 1: Please mention the lengths of the dsRNAs you used.

Line 126-128: Please make a brief introduction of your RNAi method.

Part 3.1: I could not find any figures regarding this part.

Figure 1B: Did you check the expression levels on more tissues and organisms?

Figure 2B & 2C: no significance?

Discussion: It seems that the fat body is the major expression place of the NlITIH4, so how could the fat body-based protein affect ovary? 

Author Response

Thank you for your comments and suggestions !

Line 39: Authors used a whole paragraph to make a brief introducing of N. lugens, and the conclusion of this paragraph is "These make it difficult to effectively control N. lugens", which is a big confusion to me. Because I could not find any clues on pest control aspects in this manuscript.

Response : Thanks for your comments. We have revised this paragraph for better understanding in the text (Line 37-41).

Line 68: Authors mentioned that "very little research has been done on ITIH in insects" which also means that there could still be a few studies referring to this gene. If yes, I strongly recommend listing all the related insect studies. If not, please make a proper change to this sentence.

Response : Thanks. No research has been on ITIH in insects. We have made a proper change to this sentence in the text (Line 72).

Line 69-76: The entire paragraph is a big mess, please re-write this paragraph and make a brief review on your past works. And you need to quote the articles if you have already published those data. And if the data haven't been published, I guess you could easily make a combination with the data here. 

Response : Thanks. We have written this paragraph in the text (Line 74-79).

Line 79-81: Please provide more information on the insect background and the rearing methods.

Response : Thanks. We have added more relevant information in the text (Part 2.1). “The N. lugens population used in this study was originally collected from a rice field in Yuyao (Zhejiang province, China), and the insects have been maintained in a climatron at China Jiliang University for 10 years”.

Part 2.2: How did you get the full CDS information of the NlITIH4? By conducting RACE PCR or directly deriving from the genome database?

Response : The NlITIH4 sequence information came from our transcriptome data. We have added this description in the “Materials and Methods” (Part 2.2, line 102).

Table 1: Please mention the lengths of the dsRNAs you used.

Response : We have described the length information in Table 1.

Line 126-128: Please make a brief introduction of your RNAi method.

Response : We have described more information about RNAi method in the “2.4. Double strand RNA Synthesis and Injection” ( Line 140-143).

Part 3.1: I could not find any figures regarding this part.

Response : Thank you. We have added relevant figures (Figure 1-“Nucleotide and encoded amino acid sequences of NlITIH4 ORF” and Figure 2-“A phylogenetic analysis of ITIH4 based on amino acid sequence” ).

Figure 1B: Did you check the expression levels on more tissues and organisms?

Response :. During dissection, after we took out the ovaries, fat bodies and guts, the residual insect bodies were also used to check the expression of ITIH4, and the expression level was close to that in fat body.

Figure 2B & 2C: no significance?

Response :  Yes, in 3.3 we have described the significance. About (old) Figure 2C, we revised it (new Figure 4C).

Discussion: It seems that the fat body is the major expression place of the NlITIH4, so how could the fat body-based protein affect ovary? 

Response : We do not know the exact reason, since there is no report on insects ITIH4. Maybe it can be transported to the ovary or affect some molecular that interacted with ITIH4.

Reviewer 4 Report

The manuscript by Ji et al. describes the cloning and functional role of the inter-alpha-trypsin inhibitor heavy chain 4 (ITIH4) gene in the brown planthopper, Nilaparvata lugens. Using bioinformatics tools, the authors describe the structure of the predicted protein, and using RNAi, they show that the gene has a role on oogenesis. Knockdown of the ITIH4 not only had an impact on egg production, but also insect viability. The authors also noted that knockdown of the ITIH4 gene’s transcripts resulted in reduced vitellogenin and vitellogenin receptor gene expressions. The authors provide some possible mechanisms by which ITIH4 can affect Vg uptake in the ovaries, oocyte development, and overall insect reproductive function. The methods were sufficiently well described, and the results were generally clearly presented. The discussion offered some intriguing suggestions on the molecular roles of ITIH4 in this insect. The manuscript will be of interest to researchers studying insect female oogenesis, and is worth publishing. There are, however, some questions that need to be address first:

Lines 150- 155. The authors describe the protein’s predicted structure in words, but a figure showing the various functional domains would be helpful. Does this insect inter-alpha-trypsin inhibitor heavy chain protein have all of the functional domains typically seen in this protein from mammalian systems?

Lines 155-158. The authors describe some phylogenetic analyses, but no figure (phylogenetic tree) or data (sequence comparison table) are provided. How similar (i.e. % identity) is the N. lugens proteins with that of other insects? How well defined are the insect orthologues? There do not seem to be any published reports on insect ITIH4 genes/proteins. Accession numbers should be provided.

Lines 108-108. Why were 18S rRNA transcripts used as the internal reference of the QRT-PCR analyses? Ribosomal RNAs typically outnumber single copy genes’ transcripts levels by multiple orders of magnitude.

Figure 1. Define the y-axis of Fig 1a and b. Relative to the 18S rRNA transcript levels? If so, the values seem unbelievably high, as ribosomal RNA is typically extremely abundant. This would suggest that ITIH4 transcripts are 3 to 15 times higher than 18S transcript levels!

Throughout the Results and Discussion, the authors refer to the role of the ITIH4 gene in affecting ovarian development. They do not mean “ovarian” development, but “oogenesis”.

The Introduction describes the role of ITIH4 in mammalian systems, but the authors do not indicate that this protein has not been described before in insects. This point is worth making, as this is a novel aspect of their study.

Author Response

Thank you for your comments and suggestions !

Lines 150- 155. The authors describe the protein’s predicted structure in words, but a figure showing the various functional domains would be helpful. Does this insect inter-alpha-trypsin inhibitor heavy chain protein have all of the functional domains typically seen in this protein from mammalian systems? Lines 155-158. The authors describe some phylogenetic analyses, but no figure (phylogenetic tree) or data (sequence comparison table) are provided. How similar (i.e. % identity) is the N. lugens proteins with that of other insects? How well defined are the insect orthologues? There do not seem to be any published reports on insect ITIH4 genes/proteins. Accession numbers should be provided.

Response : Thank you for your comments. ITIH4 of mammals (like Homo sapiens, Sus scrofa and Mus pahari, etc.) also contain the two functional domains (VIT, vWFA) that exist in insects. Again,We blasted ITIH4 homologues from all other insects that are available in NCBI, even though there is no report about insect ITIH4 yet. ITIH4 from Homalodisca vitripennis had the highest similarity (48.56%) with NlITIH4. For better understanding,we added the sequence of N. lugens ITIH4 and a phylogenetic tree of ITIH4 was also showed. (new Figure 1 and Figure 2)

Lines 108-108. Why were 18S rRNA transcripts used as the internal reference of the QRT-PCR analyses? Ribosomal RNAs typically outnumber single copy genes’ transcripts levels by multiple orders of magnitude. Figure 1. Define the y-axis of Fig 1a and b. Relative to the 18S rRNA transcript levels? If so, the values seem unbelievably high, as ribosomal RNA is typically extremely abundant. This would suggest that ITIH4 transcripts are 3 to 15 times higher than 18S transcript levels!

Response : We have considered several candidate reference genes,including 18S,GADPH and beta-actin, etc. Indeed, the expression of 18S is much higher that target genes, but considering its stability, we took it as the reference gene. When we did qPCR, we diluted the cDNA 10-fold.

According to 2−∆∆Ct method (Livak,K.J.; Schmittgen,T.D. Analysis of Relative Gene Expression Data Using Real-Time Quantitative PCR and the 2−ΔΔCT Method. Methods. 2001, 25, 402-408. doi:10.1006/meth.2001.1262), when we did relative expression analysis, and set the expression level of the sample who had the highest value of [CT(target gene) - CT(18S)] as 1. We were also surprised by the high efficiency of RNAi to the expression of ITIH4. Maybe this is the reason why the dsITIH4-treated insects died so quickly from the 1d.p.i. compared with dsGFP-treated.

Throughout the Results and Discussion, the authors refer to the role of the ITIH4 gene in affecting ovarian development. They do not mean “ovarian” development, but “oogenesis”.

Response : Oogeneis is the most important part during ovarian development. In this study, we described more about appearance change of the ovarioles or whole ovaries. If we focus on “oogenesis”, we need to show more details. Besides the ovarian development, we also considered the survival, oviposition and hatchability. Therefore, we say that NlITIH4 plays an important role in the development and reproduction. Thank you for your comment again!

The Introduction describes the role of ITIH4 in mammalian systems, but the authors do not indicate that this protein has not been described before in insects. This point is worth making, as this is a novel aspect of their study.

Response : Thank you for your comments. ITIH4 has not been described before in insects. We have revised the relevant information in introduction (Line 72).

Reviewer 5 Report

The manuscript by Ji et al investigated the function of NlITIH4 gene in development and reproduction of the female brown planthopper by using RNAi, and found that NlITIH4 plays a role in insect survival, ovarian development and oocyte maturation as well as egg production and hatchability. These data might be useful for biological control of this destructive pest.

Some concerns and comments:

Line 69, In previous studies (?), please add references.

In results section 3.1, it is better to provide a figure including the gene structure and the protein domains, a phylogenetic tree (NlITIH4 orthologs from other species). In present version of the manuscript, no information is found of which insects the author used for sequence comparison or alignment.

Line 181 and 182, “The dsNlITIH4 treatment group” change into “The dsNlITIH4-treated group”.

Rewrite the paragraph from line 180 to line 184 make it clear!    

Line194, “dsGFP4-treated?” should be “dsGFP-treated”

Line 195, delete the “went to late or terminal vitellogenesis”

The caption in Figure 3 is too short!

In Figure 4, the expression trend of Vg/VgR was weird. The transcript level of Vg increased from 1 dpi to 4 dpi in the dsGFP-treated group, but the VgR decreased.  Please see reference 31, both genes were increasingly expressed from 1-7 days after female adult emergence.

What kind of cDNA templates used for the qRT-PCR in Figure 4?

Line 282, “data not shown”, please provide data!

NlITH4 was also highly expressed in the gut of the insect, I just wonder whether there are some gut defects, or dieting behavior changes in the dsNlITH4-injected insects?

(Kang et al. 2022, Effects of Different Nutritional Conditions on the Growth and Reproduction of Nilaparvata lugens (Stål). DOI: 10.3389/fphys.2021.794721)

Author Response

Thank you for your comments and suggestions !

Line 69, In previous studies (?), please add references.

Response : Thank you for your comment. We have revised this description to “experiment”(Line 74).

In results section 3.1, it is better to provide a figure including the gene structure and the protein domains, a phylogenetic tree (NlITIH4 orthologs from other species). In present version of the manuscript, no information is found of which insects the author used for sequence comparison or alignment.

Response : We are grateful for your suggestion. We have added the relevant figures about ITIH4 sequence and phylogenetic tree in the text (Part 3.1).

Line 181 and 182, “The dsNlITIH4 treatment group” change into “The dsNlITIH4-treated group”.

Response : Thank you for your careful review. We have revised them.  (Part 3.3, line 249).

Rewrite the paragraph from line 180 to line 184 make it clear!    

Response : To be more clear, we have rewritten this sentence in the text (Line 249-253).

Line194, “dsGFP4-treated?” should be “dsGFP-treated”

Response : Thank you for your careful review. We revised them. (Line 263).

Line 195, delete the “went to late or terminal vitellogenesis”

Response : Thanks . We have revised this sentence (Line 264).

The caption in Figure 3 is too short!

Response : We have added more information in Figure 5 (old Figure 3).

In Figure 4, the expression trend of Vg/VgR was weird. The transcript level of Vg increased from 1 dpi to 4 dpi in the dsGFP-treated group, but the VgR decreased.  Please see reference 31, both genes were increasingly expressed from 1-7 days after female adult emergence.

Response : We are extremely grateful to you for pointing out this problem. We re-analyzed all qPCR data, and we found some mistakes during calculation. We re-drew the figure, and the results make sense now (Figure 6). Thanks again for point this mistake.

What kind of cDNA templates used for the qRT-PCR in Figure 4?

Response : Total RNA from entire insect body. We described it in the text (Part 2.5, line 152-155).

Line 282, “data not shown”, please provide data!

Response : We have shown the relevant data in Figure 6C (about Met expression) and revised relevant discussion in the text (Line 357-365).

NlITH4 was also highly expressed in the gut of the insect, I just wonder whether there are some gut defects, or dieting behavior changes in the dsNlITH4-injected insects? (Kang et al. 2022, Effects of Different Nutritional Conditions on the Growth and Reproduction of Nilaparvata lugens (Stål). DOI: 10.3389/fphys.2021.794721).

Response : In this study, we also observed the phenotype of gut under a microscope after RNAi, but we did not found some obvious difference. We will try to use other methods to find the difference. If ITIH4 affect dieting behavior of N. lugens, this should be an interesting work. Thanks for introducing us the paper again, anyway.

Round 2

Reviewer 1 Report

The revised manuscript looks good. All comments and suggestions have been revised in the updated manuscript.

Reviewer 3 Report

My concerns have been addressed, thanks.

Reviewer 4 Report

The authors have adequately addressed my previous concerns with the manuscript. I think the paper is suitable for publication.